IFT-UAM/CSIC-24-093
arXiv:2408.04621 [gr-qc]
August 6ᵗʰ, 2024

# On the interactions and equilibrium between Einstein-Maxwell-Dilaton black holes

*Ulrich K. Beckering Vinckers*[1,2,a] *and Tomás Ortín*[3,b]

[1]*Cosmology and Gravity Group, Department of Mathematics and Applied Mathematics, University of Cape Town, Rondebosch 7701, Cape Town, South Africa*

[2]*Van Swinderen Institute, University of Groningen, 9747 AG Groningen, The Netherlands*

[3]*Instituto de Física Teórica UAM/CSIC C/ Nicolás Cabrera, 13–15, C.U. Cantoblanco, E-28049 Madrid, Spain*

## Abstract

We study the interactions and the conditions for the equilibrium of forces between generic non-rotating black holes of the Einstein–Maxwell–Dilaton (EMD) theory. We study known (and some new) solutions of the time-symmetric initial-data problem describing an arbitrary number of those black holes, some of them with primary scalar hair. We show how one can distinguish between initial data corresponding to dynamical situations in which the black holes (one or many) are not in equilibrium and initial data which are just constant-time slices of a static solution of the full equations of motion describing static black holes using (self-)interaction energies. For a single black hole, non-vanishing self-interaction energy is always related to primary scalar hair and to a dynamical black hole. Removing the self-interaction energies in multi-center solutions we get interaction energies related to the attractive and repulsive forces acting on the black holes. As shown by Brill and Lindquist, for widely separated black holes, these take the standard Newtonian and Coulombian forms plus an additional interaction term associated with the scalar charges which is attractive for like charges.

---
[a]Email: `bckulr002[at]myuct.ac.za`
[b]Email: `tomas.ortin[at]csic.es`

# 1  Introduction

The study of the general-relativistic description of the gravitational interaction between massive bodies and their motion beyond the idealized test-particle limit is one of the most important research subjects in this field, from the pioneering works of Einstein, Grommer, Infeld, Hoffman, and Papapetrou among others [1–6] all the way to the recent detection of the gravitational radiation produced in the gravitational interaction of two black holes [7].

A very interesting approach to this problem is the study of particular solutions to the initial-data problem that describe more than one gravitating object (black holes, in general). This is a non-trivial problem which contains a substantial deal of information about the interactions between the black holes. In particular, as shown by Brill and Lindquist in Refs. [8, 9], one can find the forces between the back holes and the conditions for their equilibrium recovering, for large separations, the Newtonian limit of the forces. Brill and Lindquist's approach makes use of the great simplifications taking

place in the so-called time-symmetric case[1] in which the Cauchy surface on which the initial-date problem is formulated corresponds to a moment in which the black holes are instantaneously at rest, but, usually, subject to non-vanishing accelerations. Time-symmetric initial data can be found by analytical methods, as we are going to see, but they can only describe head-on collisions of non-rotating black holes as discussed in Ref. [10]. The time-symmetric initial data obtained by Misner in Ref. [11] and which are particular cases of those obtained by Brill and Lindquist, have been used to study head-on collisions of Schwarzschild black holes in Refs. [12–14], for instance.

In Ref. [16] Brill and Lindquist's study was extended to theories including a scalar field. Some of the results obtained, corroborated and further extended in Ref. [17], were quite unusual, as we are going to explain

In this approach, each black hole is associated with an Einstein–Rosen-like bridge connecting two asymptotically-flat regions [22]. There is one asymptotically-flat region to which all the bridges are connected and which corresponds to "our universe" in which all the black holes are present at a given time and each bridge leads to another universe in which only one black hole is present. One can compute the total mass of the set of interacting black holes in our universe and the mass of each individual black hole in each of the single-black-hole universes. The total mass does not equal the sum of the individual masses and the difference can be interpreted as the interaction energy between the black holes, from which the force between two black holes can be extracted. For single black holes of the Einstein–Maxwell theory considered by Brill and Lindquist in Refs. [8, 9], the masses computed on both universes are always identical and the interaction energies vanish, but in some of the solutions found in Refs. [16, 17] this was no longer true: there are single black-hole initial data connecting two universes in which the black hole seems to have different mass, scalar charge and asymptotic value of the scalar. To the best of our knowledge, no physical interpretation of this fact has been given so far.

In this paper we want to revisit this puzzle, put forward an explanation and test it in different solutions of the time-symmetric initial-data problem of the Einstein-Maxwell-Dilaton system, some of them, new.[2] We are going to argue that the difference in the values of the masses and scalar charges in two universes is associated with the non-staticity of the complete solution because, for static solutions, one can construct a Komar charge [23–26] and a scalar charge [27, 28] that satisfy Gauss-type laws.[3] This

---

[1]See, *e.g.* Ref. [15].

[2]The interactions between black-hole solutions of this system, specially, between the dyonic ones, have been recently studied in Ref. [18].

On the other hand, the time evolution and collisions of Einstein-Maxwell-Dilaton black holes with more generic initial data have been studied by numerical methods in Refs. [19–21]. The methods and goals of those works are quite different from those of this paper.

[3]These charges are written as $(d-2)$-forms in $d$ dimensions and one can show that they are on-shell closed if the spacetime admits a Killing vector. The closedness of the $(d-2)$-form charge is equivalent to the Gauss law. In our particular case, integrating the exterior derivative of the Komar charge (a 3-form that vanishes on shell) over the 3-dimensional Cauchy surface and applying Stokes theorem, we find that the sum of the integrals of the Komar charge 2-form over all the 2-dimensional boundaries (2-spheres at

implies that the mass and scalar charge must necessarily be the same in both sides of the Einstein–Rosen bridge when the total spacetime is static. Then, the difference in masses and scalar charges implies the non-staticity of the complete solution. The initial data correspond to a dynamical black hole which is evolving and which, furthermore, has primary scalar hair.[4] The evolution of a black hole which has primary scalar hair is a very interesting problem related to the weak cosmic censorship and no-hair conjectures which deserves further study but falls outside the scope of this work.

This paper is organized as follows: in Section 2 we present the Einstein–Maxwell-Dilaton theory in our conventions. In Section 3 we formulate the time-symmetric initial data problem for this theory and in Section 4 we present and study different solutions of increasing complexity: in Section 4.1 we review solutions with vanishing scalar, electric and magnetic fields, showing how they describe interacting Schwarzschild black holes. In Section 4.2 we include a non-trivial scalar field and we find solutions describing regular black holes, with non-vanishing interaction energy even for the single black-hole case (*self-interaction energy*). When this self-interaction energy vanishes, we show that the solutions are constant-time slices of the (singular) Janis–Newman–Winicour solutions [30]. In Section 4.3 we consider non-trivial electric fields but vanishing scalar (the Einstein–Maxwell theory), recovering the results obtained by Brill and Lindquist in Ref. [9]. In Section 4.4 we describe how to generalize the purely electric solutions of the previous section to purely magnetic or dyonic ones. In Section 4.5 we consider the solutions of Ref. [17], which have non-trivial electric and scalar fields. We show how, when the self-interaction energies vanish, the solutions describe constant-time slices of the well-known static, regular, electric dilaton black-hole solutions of Refs. [31–33] which have no primary scalar hair. Then, eliminating these self-interaction energies, we can determine the interaction energy and force between two of these black holes, recovering the expected result. We generalize the results of this section to the dyonic $a = 1$ case in Section 4.6. We discuss our results and possible future directions in Section 5. Finally, the appendices contain some of the complete spacetime solutions mentioned in the main text and their transformation to coordinates that allow us to get constant-time slices at the moment of time-symmetry showing the two sides of the Einstein–Rosen bridge.

---

infinity) must vanish. Each integral gives the mass in the corresponding asymptotic region and, taking into account the orientation of each boundary we find that the mass at each side of an Einstein–Rosen bridge for a single, static, black hole must take the same value.

[4]Secondary scalar hair, which is physically allowed [29], is related to the conserved charges (mass, electric and magnetic charges) by a fixed formula [28]. In all the cases that we have studied in which the masses and scalar charges are different in the two sides of the Einstein–Rosen bridge, the scalar charge is an independent parameter or combination of parameters and this relation is not satisfied. Thus, it corresponds to primary scalar hair.

## 2  The theory

The family of theories that we are going to consider are collectively known in the literature as the Einstein-Maxwell-Dilaton (EMD) model. The action, in differential-form language, is given by[5]

$$S[e, A, \phi] = \frac{1}{16\pi} \int \left\{ - \star (e^a \wedge e^b) \wedge R_{ab} + \tfrac{1}{2} d\phi \wedge \star d\phi + \tfrac{1}{2} e^{-a\phi} F \wedge \star F \right\}, \qquad (2.1)$$

and contains the real parameter $a$ which determines the strength of the coupling of the dilaton $\phi$ (a real scalar field) to the Maxwell field $A = A_\mu dx^\mu$ with field strength $F = dA$. Different values of $a$ correspond to different theories.

The (left-hand sides of the) equations of motion are defined by the general variation of the action generated by an arbitrary variation of the fields (here $\varphi$ stands for all the fields of the theory)

$$\delta S = \int \left\{ \mathbf{E}_a \wedge \delta e^a + \mathbf{E}_\phi \delta\phi + \mathbf{E}_A \wedge \delta A + d\mathbf{\Theta}(\varphi, \delta\varphi) \right\}, \qquad (2.2)$$

and are given by

$$\mathbf{E}_a = \imath_a \star (e^b \wedge e^c) \wedge R_{bc} + \tfrac{1}{2} \left( \imath_a d\phi \star d\phi + d\phi \wedge \imath_a \star d\phi \right)$$

$$- \frac{1}{2} e^{-a\phi} \left( \imath_a F \wedge \star F - F \wedge \imath_a \star F \right), \qquad (2.3a)$$

$$\mathbf{E}_\phi = - \left( d \star d\phi + \frac{a}{2} e^{-a\phi} F \wedge \star F \right), \qquad (2.3b)$$

$$\mathbf{E}_A = -d \left( e^{-a\phi} \star F \right). \qquad (2.3c)$$

Since we are going to work with electric and magnetic fields, we have to add to these the Bianchi identity

$$\mathbf{B}_A = dF. \qquad (2.4)$$

## 3  The time-symmetric initial-data problem

Let $\hat{n} = n_\mu dx^\mu$ be the 1-form dual to the unit-norm, timelike, vector field $n = n^\mu \partial_\mu$ normal to the Cauchy surface $\Sigma$, $\imath_n \hat{n} = n^\mu n_\mu = 1$. Taking into account our mostly minus signature, the metric induced on the Cauchy hypersurface is given by

---

[5]We set $G_N^{(4)} = 1$ in this paper.

$$h = g - \hat{n} \otimes \hat{n}, \tag{3.1}$$

or, in components,

$$h_{\mu\nu} = g_{\mu\nu} - n_\mu n_\nu. \tag{3.2}$$

In order to project over the 3-dimensional surface we can use

$$1^{(3)} = 1 - \hat{n}\imath_n, \tag{3.3}$$

which is equivalent to

$$h_\mu{}^\nu = g_\mu{}^\nu - n_\mu n^\nu. \tag{3.4}$$

The extrinsic curvature is defined over the Cauchy surface as

$$K = \tfrac{1}{2}\pounds_n h. \tag{3.5}$$

In the time-symmetric case [34,8,9,15] the normal unit vector is covariantly constant on $\Sigma$

$$\nabla n \overset{\Sigma}{=} 0, \tag{3.6}$$

which implies

$$d\hat{n} \overset{\Sigma}{=} 0, \quad \Rightarrow \quad \hat{n} \wedge d\hat{n} \overset{\Sigma}{=} 0, \tag{3.7}$$

so it is hypersurface orthogonal. All this means that, if we define the time coordinate by

$$n \equiv \partial_t, \tag{3.8}$$

with $\Sigma$ defined by $t = 0$, the metric induced on $\Sigma$ is time-independent and of the form

$$ds^2 \overset{\Sigma}{=} h_{mn}dx^m dx^n, \quad m,n = 1,2,3. \tag{3.9}$$

In the time-symmetric case we also have

$$\imath_n d\phi \overset{\Sigma}{=} 0. \tag{3.10}$$

In the time-symmetric initial data problem we only need to consider the projection of the equations of motion with $n$, taking into account that the extrinsic curvature, $K$, vanishes over the surface of time symmetry because $n$ is covariantly constant there. The equations to consider are $\hat{n} \wedge n^a \mathbf{E}_a$, $\hat{n} \wedge \mathbf{E}_A$ and $\hat{n} \wedge \mathbf{B}_A$ evaluated over $\Sigma$. It is convenient to define the electric and magnetic 1-forms relative to $n$ as

$$E \equiv \imath_n F, \qquad B \equiv e^{-a\phi}\imath_n \star F. \tag{3.11}$$

The field strength and its Hodge dual can be written in terms of these 1-forms as follows:

$$F = \hat{n} \wedge E - e^{a\phi} \star (\hat{n} \wedge B),\tag{3.12a}$$

$$\star F = e^{a\phi} \hat{n} \wedge B + \star(\hat{n} \wedge E).\tag{3.12b}$$

Let us consider first the projection of the Einstein equations:

$$\hat{n} \wedge n^a \mathbf{E}_a \overset{\Sigma}{=} \hat{n} \wedge \imath_n \star (e^b \wedge e^c) \wedge R_{bc} - \tfrac{1}{2}\hat{n} \wedge d\phi \wedge \star(\hat{n} \wedge d\phi)$$

$$- \tfrac{1}{2}e^{-a\phi}(\hat{n} \wedge E) \wedge \star(\hat{n} \wedge E) - \tfrac{1}{2}e^{a\phi}(\hat{n} \wedge B) \wedge \star(\hat{n} \wedge B).\tag{3.13}$$

Next, let us consider the projection of the Maxwell equation

$$\hat{n} \wedge \mathbf{E}_A \overset{\Sigma}{=} \hat{n} \wedge d\left[e^{-a\phi} \star^{(3)} E\right],\tag{3.14}$$

where $\star^{(3)}$ is the Hodge operator on $\Sigma$ with the induced metric.

Following the same steps, the Bianchi identity can be brought to the form

$$\hat{n} \wedge \mathbf{B}_A \overset{\Sigma}{=} -\hat{n} \wedge d\left[e^{a\phi} \star^{(3)} B\right].\tag{3.15}$$

These two equations are related by electric-magnetic duality transformations that interchange $E$ and $B$ and reverse the sign of $\phi$:

$$E' = B, \qquad B' = -E, \qquad \phi' = -\phi.\tag{3.16}$$

The projection of the Einstein equations is also invariant under them.

It is now convenient to use a Vielbein basis $\{e^a = e^a{}_\mu dx^\mu\}$ in which $e^0 = \hat{n}$. The rest of the 1-forms will be denoted by $e^i$, $i = 1, 2, 3$ and, on $\Sigma$, they are a Dreibein for the 3-dimensional metric $h_{mn}$:[6]

$$ds^2 \overset{\Sigma}{=} h_{mn}dx^m dx^n \overset{\Sigma}{=} -\delta_{ij}e^i e^j, \qquad E = E_i e^i, \qquad B = B_i e^i,\tag{3.17}$$

In this basis, the above equations take the form

$$R(-h) - \tfrac{1}{2}\left[\partial_i\phi\partial_i\phi + e^{-a\phi}E_iE_i + e^{+a\phi}B_iB_i\right] = 0,\tag{3.18a}$$

$$\mathcal{D}_i\left(e^{-a\phi}E_i\right) = 0,\tag{3.18b}$$

---

[6]A more precise (but also more involved) notation would distinguish between the elements of the Vierbein $e^i$ and their pullbacks over $\Sigma$, which constitute a Dreibein of the induced metric. We will assume from this moment that all the objects and equations are 3-dimensional and defined on $\Sigma$ only.

$$\mathcal{D}_i \left( e^{a\phi} B_i \right) = 0 \,, \tag{3.18c}$$

where $R(-h)$ is the Ricci scalar of the 3-dimensional metric with positive signature.

Our ansatz for the 3-dimensional metric is always going to be of the form

$$- h_{mn} dx^m dx^n = \Phi^4 \bar{h}_{mn} dx^m dx^n \,, \tag{3.19}$$

where $\bar{h}_{mn}$ is a reference, background, 3-dimensional metric of positive signature, such as the Euclidean metric, the metric of the round $S^3$ or that of the product $S^2 \times S^1$ [17]. For this ansatz,

$$R(-h) = -\Phi^{-4} \left[ \bar{R} + 8\Phi^{-1} \bar{\nabla}^2 \Phi \right] \,, \tag{3.20}$$

where the barred objects are computed with the barred metric. Then, the above three equations can be written in the background metric in the form

$$\bar{R} + 8\Phi^{-1} \bar{\nabla}^2 \Phi + \tfrac{1}{2} \bar{h}^{mn} \left[ \partial_m \phi \partial_n \phi + e^{-a\phi} E_m E_n + e^{a\phi} B_m B_n \right] = 0 \,, \tag{3.21a}$$

$$\bar{\nabla}_m \left( \Phi^2 e^{-a\phi} E^m \right) = 0 \,, \tag{3.21b}$$

$$\bar{\nabla}_m \left( \Phi^2 e^{a\phi} B^m \right) = 0 \,. \tag{3.21c}$$

Finally, if the background metric is the 3-dimensional Euclidean metric $\mathbb{E}^3$ in Cartesian coordinates, the above equations take the simple form

$$8\Phi^{-1} \partial_m \partial_m \Phi + \tfrac{1}{2} \left[ \partial_m \phi \partial_m \phi + e^{-a\phi} E_m E_m + e^{a\phi} B_m B_m \right] = 0 \,, \tag{3.22a}$$

$$\partial_m \left( \Phi^2 e^{-a\phi} E_m \right) = 0 \,, \tag{3.22b}$$

$$\partial_m \left( \Phi^2 e^{a\phi} B_m \right) = 0 \,. \tag{3.22c}$$

This is the case that we are going to consider in this paper and these are the PDEs for which we will find solutions describing interacting black holes at the moment of time symmetry.

# 4 Solutions to the time-symmetric initial-data problem

In this section we are going to reproduce and interpret physically some well-known and some new solutions of the time-symmetric data problem equations for the EMD model, Eqs. (3.22). We will first consider the vacuum case and later we will consider the addition of matter fields of the EMD theory, gradually increasing the complexity of the solutions and introducing the concepts that we need. Most of the solutions and their generic interpretation as interacting black holes are not new. However, most of the solutions which have non-trivial scalar fields exhibit strange features (see Ref. [16]) for which we are going to propose a physical interpretation related to the static or dynamical nature of the complete spacetime solution. This will allow us to determine the forces acting between two widely-separated EMD black holes.

## 4.1 Schwarzschild black holes

Let us first consider solutions with vanishing electric and magnetic fields and constant scalar field. The only equation left to be solved is the Laplace equation in $\mathbb{E}^3$

$$\partial_m \partial_m \Phi = 0 \,. \tag{4.1}$$

A convenient set of asymptotically-flat solutions is provided by

$$\Phi = 1 + \sum_{i=1}^{N} \frac{\Phi_i/2}{|\vec{x} - \vec{x}_i|} \,, \tag{4.2}$$

where the vectors $\vec{x}_i$ are constant.

For $N = 1$, setting $\vec{x}_1 = \vec{0}$ and using spherical coordinates

$$d\vec{x}^2 = d\rho^2 + \rho^2 d\Omega_{(2)}^2 \,, \quad \text{with} \quad \rho^2 = |\vec{x}|^2 \quad \text{and} \quad d\Omega_{(2)}^2 = d\theta^2 + \sin^2\theta d\varphi^2 \,, \tag{4.3}$$

we recover the spatial part of the Schwarzschild metric with mass $M = \Phi_1$ in isotropic coordinates, given in Eq. (B.5) which has the event horizon located at $\rho = M/2$.

In the $\rho \to 0$ limit, the 3-dimensional metric

$$\left(1 + \frac{M/2}{\rho}\right)^4 \left(d\rho^2 + \rho^2 d\Omega_{(2)}^2\right) \,, \tag{4.4}$$

becomes singular, but it can be analytically continued using $\rho' \equiv (M/2)^2/\rho$. The metric takes exactly the same form, with $\rho$ replaced by $\rho'$ and now covers another asymptotically-flat region, which corresponds to a constant-time slice of region III of the Kruskal-Szekeres spacetime. We conclude that these two charts cover the $T = 0$ (time-symmetric) slice of the Kruskal-Szekeres spacetime, which is also known as the *Einstein-Rosen bridge* [22] of a black hole of mass $M$. The "neck" of the bridge coincides

with the intersection of the $T = 0$ slice with the event horizon, which is the bifurcation sphere.

When $N > 1$ we can use spherical coordinates centered at the $i^{th}$ pole of the harmonic function $\Phi$

$$d(\vec{x} - \vec{x}_i)^2 = d\rho_i^2 + \rho_i^2 d\Omega_{(2)i}^2, \quad \text{with} \quad \rho_i^2 = |\vec{x} - \vec{x}_i|^2 \quad \text{and} \quad d\Omega_{(2)i}^2 = d\theta_i^2 + \sin^2\theta_i d\varphi_i^2,$$
(4.5)

and analytically continue the metric in the $\rho_i \to 0$ direction using a new coordinate $\rho_i' = (\Phi_i/2)^2/\rho_i$. First, in the $\rho_i \to 0$ limit

$$\Phi = \frac{\Phi_i/2}{\rho_i} + A_i + \mathcal{O}(\rho_i)$$
(4.6)

where

$$A_i \equiv 1 + \sum_{j \neq i} \frac{\Phi_j/2}{|\vec{x}_{ij}|}, \qquad \vec{x}_{ij} \equiv \vec{x}_j - \vec{x}_i,$$
(4.7)

and there is no sum over the repeated indices. In the new coordinates

$$\Phi = \frac{\rho_i'}{\Phi_i/2} \left[ \left( 1 + \frac{A_i \Phi_i/2}{\rho_i'} \right) + \mathcal{O}(1/\rho_i'^2) \right],$$
(4.8)

and the full 3-dimensional metric is

$$\left[ 1 + \frac{2 A_i \Phi_i}{\rho_i'} + \mathcal{O}(1/\rho_i'^2) \right] \left( d\rho_i'^2 + \rho_i'^2 d\Omega_{(2)i}^2 \right).$$
(4.9)

Comparing with the single black-hole case, we see that, in the $\rho_i \to 0$ limit, this metric approaches that of a constant time slice of the metric of a single Schwarzschild black hole of mass

$$M_i = \Phi_i A_i = \Phi_i + \Phi_i \sum_{j \neq i} \frac{\Phi_j/2}{|\vec{x}_{ij}|}.$$
(4.10)

This implies that the original Cartesian coordinates cover a patch of the Cauchy hypersurface with $N$ interacting Schwarzschild black holes. We can only compute their total mass in that asymptotically-flat region. Expanding the metric for large values of the three Cartesian coordinates and defining $\rho \equiv |\vec{x}|$, the 3-dimensional metric behaves as

$$\sim \left[ 1 + \frac{2 \sum_{i=1}^{N} \Phi_i}{\rho} + \mathcal{O}(1/\rho^2) \right] \left( d\rho^2 + \rho^2 d\Omega_{(2)}^2 \right),$$
(4.11)

and the total mass is

$$M = \sum_{i=1}^{N} \Phi_i \,. \tag{4.12}$$

The *interaction energy* can be defined as the difference between the the total mass and the sum of the individual masses [9]

$$M_{int} \equiv M - \sum_{i=1}^{N} M_i = -\sum_i \Phi_i \sum_{j \neq i} \frac{\Phi_j/2}{|\vec{x}_{ij}|} = -\sum_{i<j} \frac{\Phi_i \Phi_j}{|\vec{x}_{ij}|} \,. \tag{4.13}$$

For $N = 2$ we can express the integration constants $\Phi_{1,2}$ in terms of the individual masses $M_{1,2}$ and the "distance"[7] $|\vec{x}_{12}|$ exactly, but the result is not very illuminating. For large separations,

$$\Phi_{1,2} = M_{1,2} - \frac{M_1 M_2/2}{|\vec{x}_{12}|} + \mathcal{O}(1/|\vec{x}_{12}|^2) \,, \tag{4.14}$$

and the interaction energy

$$M_{int} = -\frac{M_1 M_2}{|\vec{x}_{12}|} \left[ 1 - \frac{(M_1 + M_2)/2}{|\vec{x}_{12}|} + \mathcal{O}\left(\frac{1}{|\vec{x}_{12}|^2}\right) \right] \,, \tag{4.15}$$

approaches the Newtonian value when $|\vec{x}_{12}| \to \infty$.

Although in this case this observation looks quite trivial, we would like to stress the fact that, for the single Schwarzschild black hole, the mass can be computed in both coordinate patches (that is, in both asymptotic regions) and the values obtained are identical, which we can trivially and obviously associate to the vanishing of the interaction energy for a single black hole. However, in some of the cases that we are going to consider next, the interaction energy for a single black hole (which we are going to call *self-interaction energy*) will not vanish: the masses computed in the two asymptotically-flat regions at the ends of the Einstein–Rosen bridge will be different, a fact that demands a physical interpretation.

At this point it is convenient to remember that, for each Killing vector of a given spacetime, the vacuum Einstein equations admit an on-shell closed ("conserved") 2-form charge which is the Noether charge associated with the invariance of the theory under general coordinate transformations: the Komar charge [23]. In stationary space-times, when the Killing vector is the one that generates time translations, the integral of this 2-form at spatial infinity gives the ADM mass. Since the Komar 2-form is closed, it satisfies a Gauss law and, if there are two asymptotic regions whose union is the boundary of a Cauchy surface, the integrals at both spatial infinities must shed the same value for the ADM masses (after the orientation is taken into account). Thus, the identity of the ADM masses in the two asymptotic regions is associated with the staticity of the complete solution and, in the case at hand, this fact looks trivial because

---

[7] $|\vec{x}_{12}|$ would be the distance if the metric was the Euclidean metric, which is only the case asymptotically.

we knew beforehand that the solution of the initial-data problem is just a constant-time slice of the Schwarzschild solution. This identity of the masses, however, can be used as a test of the stationarity of the full solution when we only know the initial data and single black hole initial data which have different masses in different asymptotic regions must correspond to non-stationary ("dynamical") solutions of the complete equations of motion in which the black hole will evolve towards a different, stationary solution.

It should also be mentioned that it is possible to define scalar charges that only satisfy a Gauss law in stationary spacetimes [27, 28]. In stationary, spherically-symmetric spacetimes, these charges are essentially the coefficients of the $1/r$ terms in the asymptotic expansions of the scalar fields in standard spherical coordinates. When these charges are different in the two asymptotic regions of the initial data of a single black hole, as in the case studied in Ref. [16], the complete spacetime cannot be stationary. Again, we expect the initial data to evolve towards a stationary solution.

The evolution of the initial data sets with different masses and scalar charges in different asymptotic regions may provide interesting insights on the no-hair conjecture (will primary scalar hair be present in the final state?) and on cosmic censorship (will the final state exhibit naked singularities?), but it has to be studied by other means which fall out of the scope of this paper.

Let us move to the next case, in which the phenomenon observed in Ref. [16] appears for the first time.

## 4.2 Janis–Newman–Winicour solutions

Let us now consider a non-vanishing scalar field but vanishing electric and magnetic fields. Following Ref. [16], we make the following ansatz for the metric function $\Phi$ and the scalar field $\phi$ in terms of two functions $\chi$ and $\psi$ which we want to be harmonic in 3-dimensional Euclidean space:

$$\Phi^4 = \psi^\delta \chi^\gamma, \tag{4.16a}$$

$$\phi = \phi_0 + \alpha \log \psi + \beta \log \chi. \tag{4.16b}$$

Eq. (3.22a) is solved by [16]

$$\Phi^4 = \psi^2 \chi^2 (\psi/\chi)^{2\alpha}, \tag{4.17a}$$

$$\phi = \phi_0 \pm 2\sqrt{1 - \alpha^2} \log(\psi/\chi), \tag{4.17b}$$

$$\partial_m \partial_m \psi = \partial_m \partial_m \chi = 0, \tag{4.17c}$$

where the constant $\alpha$ satisfies $|\alpha| \leq 1$ and can be taken to be positive for simplicity. A convenient choice for the harmonic functions $\psi, \chi$ which gives an asymptotically-flat metric is

$$\psi = \psi_0 \left( 1 + \sum_{i=1}^{N} \frac{\psi_i}{|\vec{x} - \vec{x}_i|} \right) , \tag{4.18a}$$

$$\chi = \chi_0 \left( 1 + \sum_{i=1}^{N} \frac{\chi_i}{|\vec{x} - \vec{x}_i|} \right) , \tag{4.18b}$$

where the constants $\psi_0, \psi_i, \chi_0, \chi_i$ are also taken to be positive to avoid singularities.

The normalization of the metric at infinity in the patch covered by the Cartesian coordinates demands

$$\chi_0 = \psi_0^{\frac{\alpha+1}{\alpha-1}} . \tag{4.19}$$

This condition makes them disappear from the metric and we can also make them disappear from the dilaton through a redefinition of the constant $\phi_0$. Hence, we will set them to 1 in what follows.

In the same patch, we can compute the asymptotic value of the dilaton $\phi_\infty$, the total mass $M$ and the total scalar charge $\Sigma$ (simply defined as the coefficient of $1/\rho$ in the asymptotic expansion of the dilaton [27, 28]):

$$\phi_\infty = \phi_0 , \tag{4.20a}$$

$$M = (1 + \alpha) \sum_{i=1}^{N} \psi_i + (1 - \alpha) \sum_{i=1}^{N} \chi_i , \tag{4.20b}$$

$$\Sigma = \pm 2 \sqrt{1 - \alpha^2} \sum_{i=1}^{N} (\psi_i - \chi_i) . \tag{4.20c}$$

Let us now consider the single center solution $N = 1$, setting $\vec{x}_1 = \vec{0}$ and defining $\rho \equiv \vec{x}$.

It is easy to see that, when $\psi_1 = -\chi_1 = \omega/4$, this solution is a constant-time slice of the static Janis–Newman–Winicour (JNW) solution [30] in isotropic coordinates, given in Eqs. (B.7), which is singular for any $\Sigma \neq 0$ and becomes Schwarzschild's for $\Sigma = 0$. For positive values of the integration constants $\chi_1, \psi_1$ we get more general solutions which look singular in the $\rho \to 0$ limit but can be analytically extended to another asymptotically-flat region using the new coordinate

$$\frac{\rho}{\psi_1^{(1+\alpha)/2}\chi_1^{(1-\alpha)/2}} = \frac{\psi_1^{(1+\alpha)/2}\chi_1^{(1-\alpha)/2}}{\rho'} \, . \tag{4.21}$$

In this new region, the metric and the dilaton take the form

$$\Phi^3 d\vec{x}^2 = \left(1 + \frac{\psi_1^\alpha \chi_1^{1-\alpha}}{\rho'}\right)^{2(1+\alpha)} \left(1 + \frac{\psi_1^{1+\alpha}\chi_1^{-\alpha}}{\rho'}\right)^{2(1-\alpha)} \left(d\rho'^2 + \rho'^2 d\Omega_{(2)}^2\right) , \tag{4.22a}$$

$$\phi = \phi_0 \pm 2\sqrt{1-\alpha^2}\log\frac{\psi_1}{\chi_1} \pm 2\sqrt{1-\alpha^2}\log\frac{\left(1 + \frac{\psi_1^\alpha \chi_1^{1-\alpha}}{\rho'}\right)}{\left(1 + \frac{\psi_1^{1+\alpha}\chi_1^{-\alpha}}{\rho'}\right)} \, . \tag{4.22b}$$

Thus, for $\chi_1, \psi_1 > 0$, this solution describes an Einstein-Rosen-type bridge potentially associated with a black hole.

The asymptotic values of the dilaton, mass and scalar charge measured in this second asymptotic region are

$$\phi_\infty' = \phi_0 \pm 2\sqrt{1-\alpha^2}\log\frac{\psi_1}{\chi_1} \, , \tag{4.23a}$$

$$M' = (1+\alpha)\psi_1^\alpha \chi_1^{1-\alpha} + (1-\alpha)\psi_1^{1+\alpha}\chi_1^{-\alpha} \, , \tag{4.23b}$$

$$\Sigma = \pm 2\sqrt{1-\alpha^2}\left(\psi_1^\alpha \chi_1^{1-\alpha} - \psi_1^{1+\alpha}\chi_1^{-\alpha}\right) , \tag{4.23c}$$

and are, generically, different from those computed in the first asymptotic region [16].

According to the discussion at the end of the preceding section, this difference is due to the non-stationary nature of the full spacetime corresponding to these initial data. Therefore, these initial data describe black holes which will evolve, settling to the only known static solutions of this theory: the JNW solutions which include the Schwarzschild black hole for vanishing scalar charge. In its evolution the black hole may lose all its scalar charge and end up as a Schwarzschild black hole, with a singularity hidden under an event horizon, or may fail to do it, ending up as a singular JNW solution, violating the weak cosmic censorship conjecture. This is clearly an important issue that deserves further investigation.

Finally, observe that, for $\alpha = 0$, with $\chi_1 = 0$ (the case $\psi_1 = 0$ is equivalent), the $\rho \to 0$ limit of the solution can be brought to the form

$$\Phi^4 d\vec{x}^2 \sim \psi_1^2 dx^2 + \psi_1^2 d\Omega_{(2)}^2 \, , \tag{4.24a}$$

$$\phi \sim \mp 2x\,, \tag{4.24b}$$

which describes an infinite cylindrical throat with spherical ($S^2$) section of radius $\psi_1$ and with a linear dilaton. In this region we cannot compute the mass in the standard fashion. The infinite, cylindrical throat is characteristic of extremal, static black holes and its occurrence in a solution of this kind is new. It is unclear to which solution of the full 4-dimensional system it corresponds to.

## 4.3 Electric Reissner–Nordström black holes

Next, we are going to consider charged solutions of the Einstein–Maxwell (EM) theory, which correspond to the $a = 0$ case of Eq. (2.1) with constant scalar (which we are going to ignore).

In this case, following Ref. [9], we make an ansatz based on 2 would-be harmonic functions:

$$\Phi^4 = \sigma^\delta \kappa^\gamma\,, \tag{4.25a}$$

$$E_m = \partial_m \left( \alpha \log \sigma + \beta \log \kappa \right)\,. \tag{4.25b}$$

This ansatz leads exactly to the same form of Eq. (3.22a) as the ansatz for the JNW solutions and, therefore, admits the same solutions. It is not difficult to see that Eq. (3.22b) forces the parameter that we called $\alpha$ in the previous case to vanish. Thus, we arrive at the following solutions of the initial data problem [9]

$$\Phi^4 = \sigma^2 \kappa^2\,, \tag{4.26a}$$

$$E_m = 2\partial_m \log \left( \sigma / \kappa \right)\,, \tag{4.26b}$$

$$\partial_m \partial_m \sigma = \partial_m \partial_m \kappa = 0\,. \tag{4.26c}$$

The electrostatic potential $P$ is defined by

$$E_m = \partial_m P\,, \tag{4.27}$$

up to an additive constant that should be physically irrelevant. We can take

$$P = 2 \log \left( \sigma / \kappa \right)\,. \tag{4.28}$$

Again, a convenient choice for the harmonic functions $\sigma, \kappa$ that gives an asymptotically-flat metric with the standard normalization is

$$\sigma = 1 + \sum_{i=1}^{N} \frac{\sigma_i}{|\vec{x} - \vec{x}_i|} \, , \qquad \text{(4.29a)}$$

$$\kappa = 1 + \sum_{i=1}^{N} \frac{\kappa_i}{|\vec{x} - \vec{x}_i|} \, , \qquad \text{(4.29b)}$$

where the integration constants $\sigma_i$ and $\kappa_i$ have to be positive to avoid singularities. It is not too difficult to see that, for a single center ($N = 1$) these solutions are just constant-time slices of the Reissner–Nordström (RN) solutions given in Eqs. (B.13) and (B.14).[8]

The total mass, electric charge and asymptotic value of the electrostatic potential Eq. (4.28) of these solutions computed in the asymptotically-flat patch covered by these coordinates are

$$M = \sum_{i=1}^{N} (\sigma_i + \kappa_i) \, , \qquad \text{(4.31a)}$$

$$q = 2 \sum_{i=1}^{N} (\sigma_i - \kappa_i) \, , \qquad \text{(4.31b)}$$

$$P_\infty = 0 \, . \qquad \text{(4.31c)}$$

Observe that the total mass and charge satisfy the bound

$$M \geq |q|/2 \, . \qquad \text{(4.32)}$$

Let us analyze the $\rho_i \equiv |\vec{x} - \vec{x}_i| \to 0$ limit in which the metric seems to be singular. If either $\sigma_i$ or $\kappa_i$ vanishes, the metric approaches the cylindrical metric Eq. (4.24a) characteristic of extremal black holes. If neither of them vanishes, the metric is singular in these coordinates and we must analytically continue it using

$$\frac{\rho_i}{(\sigma_i \kappa_i)^{1/2}} = \frac{(\sigma_i \kappa_i)^{1/2}}{\rho'_i} \, . \qquad \text{(4.33)}$$

---

[8]At first sight, the RN solutions depend on three functions. However, the identity $\alpha(2\psi_1 - \sigma_1 - \kappa_1) = 2(\sigma_1 - \kappa_1)$ allows us to rewrite the constant-time slices (in particular, the electric field) entirely in terms of just $\sigma$ and $\kappa$. Observe that

$$\Phi^2 E_m = \sqrt{|g|} g^{tt} g^{mn} \partial_n A_t \, , \qquad \text{(4.30)}$$

where the right-hand side of the equation is computed using the fields of the complete spacetime solution.

In the new coordinates that cover the $i^{\text{th}}$ asymptotically-flat region, the metric, the electric field and the electrostatic potential (with the same choice of additive constant as in Eq. (4.28)) take the form

$$ds_\beta^2 \sim \left(1 + \frac{\kappa_i A_i}{\rho_i'}\right)^2 \left(1 + \frac{\sigma_i B_i}{\rho_i'}\right)^2 \left(d\rho_i'^2 + \rho_i'^2 d\Omega_{(2)\,i}^2\right), \tag{4.34a}$$

$$E_{m_i'} \sim 2\partial_{m_i'} \log\left[\left(1 + \frac{\kappa_i A_i}{\rho_i'}\right)\left(1 + \frac{\sigma_i B_i}{\rho_i'}\right)^{-1}\right], \tag{4.34b}$$

$$P \sim 2\log\left[\left(1 + \frac{\kappa_i A_i}{\rho_i'}\right)\left(1 + \frac{\sigma_i B_i}{\rho_i'}\right)^{-1}\right], \tag{4.34c}$$

$$A_i = 1 + \sum_{j \neq i} \frac{\sigma_i}{|\vec{x}_{ji}|}, \tag{4.34d}$$

$$B_i = 1 + \sum_{j \neq i} \frac{\kappa_i}{|\vec{x}_{ji}|}, \tag{4.34e}$$

and we can compute the masses, electric charges and electrostatic potentials at the $i^{th}$ asymptotic region[9]

$$M_i = \sigma_i + \kappa_i + \sum_{j \neq i} \frac{\sigma_i \kappa_j + \kappa_i \sigma_j}{|\vec{x}_{ij}|}, \tag{4.35a}$$

$$q_i = 2\left[\sigma_i - \kappa_i + \sum_{j \neq i} \frac{\sigma_i \kappa_j - \kappa_i \sigma_j}{|\vec{x}_{ij}|}\right], \tag{4.35b}$$

$$P_{\infty_i} = 0, \tag{4.35c}$$

and the bound

---

[9]The sign of the charges computed in the $i^{th}$ asymptotic regions is the opposite if we take the orientation that woud be outward-pointing in that region. We have taken the sign that corresponds to the opposite orientation. Since the sign of the charge is purely conventional, only the relative signs of the charges at both sides of the Einstein–Rosend bridges are relevant. Our choice is such that the sum of the charges at both sides are equal. The opposite choice would give a vanishing "total charge".

$$M_i \geq |q_i|/2, \tag{4.36}$$

is also satisfied in the $i^{th}$ asymptotic region. It is easy to see that it is saturated in the limit in which either $\kappa_i$ or $\sigma_i$ vanishes.

The total charge computed in the first asymptotic region, $q$, is equal to the sum of these charges

$$q = \sum_{i=1}^{N} q_i, \tag{4.37}$$

because the electric field satisfies a Gauss law Eq. (3.22b). The energy does not satisfy a Gauss law and the interaction energy

$$M_{int} = M - \sum_{i=1}^{N} M_i = -2 \sum_{i \neq j} \frac{\sigma_i \kappa_j}{|\vec{x}_{ij}|}, \tag{4.38}$$

does not vanish in general, except for the single-center case.

We can solve for the integration constants for large separations $|\vec{x}_{ij}|$ in terms of the physical parameters $M_i, q_i$

$$\sigma_i \sim \tfrac{1}{2}(M_i + q_i/2) \left[ 1 - \tfrac{1}{2} \sum_{j \neq i} \frac{M_j - q_j/2}{|\vec{x}_{ij}|} + \mathcal{O}\left(1/|\vec{x}_{ij}|^2\right) \right], \tag{4.39a}$$

$$\kappa_i \sim \tfrac{1}{2}(M_i - q_i/2) \left[ 1 - \tfrac{1}{2} \sum_{j \neq i} \frac{M_j + q_j/2}{|\vec{x}_{ij}|} + \mathcal{O}\left(1/|\vec{x}_{ij}|^2\right) \right], \tag{4.39b}$$

and express the interaction energy as

$$M_{int} = -\sum_{i<j} \frac{M_i M_j}{|\vec{x}_{ij}|} + \sum_{i<j} \frac{q_i q_j/4}{|\vec{x}_{ij}|} + \mathcal{O}\left(1/|\vec{x}_{ij}|^2\right), \tag{4.40}$$

*i.e.* as the sum of an attractive Newtonian contribution and a repulsive (for like charges) Coulombian contribution.

The total interaction energy vanishes when

$$M_i = |q_i|/2, \quad \forall i = 1, \cdots, N, \tag{4.41}$$

which implies that either $\sigma_i = 0$ or $\kappa_i = 0$. As we have seen above, in this case we cannot really define the individual masses $M_i$ because the region described by the new coordinates is not asymptotically flat, but cylindrical. However, it is natural to accept that they approach the extremal value $|q_i|/2$.

We would like to associate a non-vanishing interaction energy to a non-equilibrium situation as in the previous examples. This is what the results seem to indicate, but in this case we have to take into account that the standard Komar charge is not closed on-shell and the mass does not satisfy a Gauss law in stationary EM spacetimes. There is, however, an on-shell closed *generalized Komar charge* [24–26] whose integral at spatial infinity gives the combination $M - P_\infty q$. Since, in this case, the electrostatic potential takes the same (vanishing) value in all the spatial infinities and the electric charge always satisfy a Gauss law, it follows that the mass must also satisfy a Gauss law and, when it does not (*i.e.* when there is a non-vanishing interaction energy) the complete spacetime cannot be stationary and must evolve towards a stationary spacetime which should be a RN black hole.

In contrast with the Einstein-Dilaton case, single-center solutions always have vanishing self-interaction energy, at least within the class described by our ansatz. As we have mentioned, they are constant-time slices of the static RN solutions.

## 4.4 Magnetic and dyonic Reissner-Nordström black holes

Magnetic solutions can be obtained by simply using the electric-magnetic duality transformations Eqs. (3.16). In the particular case of the Maxwell theory ($a = 0$) those discrete transformations are a particular case ($\alpha = \pi/2$) of the continuous group of electric-magnetic duality transformations

$$E' = \cos \alpha \, E + \sin \alpha \, B\,, \qquad B' = -\sin \alpha \, E + \cos \alpha \, B\,, \tag{4.42}$$

which can be used to generate dyonic solutions out of the purely electric ones. These transformations leave the metric invariant and the solutions have the same properties as the purely electric ones with the electric charges $q$ replaced by $\sqrt{q^2 + p^2}$.

## 4.5 Electric EMD black holes

In Ref. [17], Cvetič, Gibbons and Pope found a solution to Eqs. (3.22a) and (3.22b) with a non-trivial dilaton field based on three harmonic functions, generalizing the solution found in Ref. [16].

Based on our experience with the RN solution, we could simply make an ansatz based on a constant-time slice of the well-known electric, static, spherically-symmetric, black-hole solutions of the EMD model (reviewed in Appendix A) written in isotropic coordinates. These solutions are given in Eqs. (B.9), (B.10) and (B.11). The full metric depends on 4 functions, but the spatial part, the vector field and the dilaton field depend on just three: $\sigma, \kappa, \psi$. Furthermore, just as in the RN case, the relations between the integration constants allow us to rewrite the electric field in terms of just two, $\sigma$ and $\kappa$. Thus, we can directly try the ansatz

$$\Phi^4 = (\sigma\kappa)^{\frac{2}{1+a^2}}\psi^{\frac{4a^2}{1+a^2}}, \tag{4.43a}$$

$$E_m = \frac{2}{\sqrt{1+a^2}}e^{a\phi/2}\partial_m \log\left(\sigma/\kappa\right), \tag{4.43b}$$

$$e^{-a\phi} = e^{-a\phi_\infty}\left(\frac{\sigma\kappa}{\psi^2}\right)^{\frac{2a^2}{1+a^2}}, \tag{4.43c}$$

and we find that it satisfies Eqs. (3.22a) and (3.22b) when $\psi, \sigma, \kappa$ are arbitrary harmonic functions in 3-dimensional Euclidean space $\mathbb{E}^3$

$$\partial_m\partial_m(\psi, \sigma, \psi) = 0, \tag{4.44}$$

where, as usual, we assume that the integration constants $\psi_i, \sigma_i, \kappa_i$ are not negative.

These are the solutions of the EMD initial data problem found in Ref. [17].

Now we choose the harmonic functions

$$\psi, \sigma, \kappa = 1 + \sum_{i=1}^{N} \frac{\psi_i, \sigma_i, \kappa_i}{|\vec{x} - \vec{x}_i|}, \tag{4.45}$$

in the obvious notation.

The total mass, electric and scalar charges of these solutions computed in the asymptotically-flat patch covered by these coordinates are[10]

$$M = \frac{1}{1+a^2}\sum_{i=1}^{N}\left(\sigma_i + \kappa_i + 2a^2\psi_i\right), \tag{4.47a}$$

$$q = \frac{2e^{-a\phi_\infty/2}}{\sqrt{1+a^2}}\sum_{i=1}^{N}\left(\sigma_i - \kappa_i\right). \tag{4.47b}$$

---

[10]The electric charge can be defined in a coordinate-invariant way by

$$q = \frac{1}{4\pi}\int_{S^2} d^2\Sigma_m e^{-a\phi}E^m = -\rho^2\Phi^2 e^{-a\phi}E_\rho. \tag{4.46}$$

Since this charge satisfies a Gauss law, this expression must be $\rho$-independent.

There is no similar definition for the scalar charge, in general, and, as it is customary [35], we define it as the coefficient of $1/\rho$ in the asymptotic expansion of the dilaton.

$$\Sigma = -\frac{2a}{1+a^2} \sum_{i=1}^{N} (\sigma_i + \kappa_i - 2\psi_i) \,. \tag{4.47c}$$

Before studying the generic $\rho_i \equiv |\vec{x} - \vec{x}_i| \to 0$ limit let us consider that limit in the single center case, choosing $\vec{x}_1 = 0$ for simplicity. Observe that this solution has 3 independent integration constants, $\psi_1, \sigma_1, \kappa_1$, that describe 3 independent physical constants (apart from $\phi_\infty$): $M, q, \Sigma$. Therefore, these solutions have primary scalar hair described by $\Sigma$.

When the three integration constants are different from zero, $\Phi^4 \sim 1/\rho^4$ in that limit, and we can analytically extend the metric using the coordinate $\rho'$ defined by

$$\frac{\rho'}{\Omega^{1/2}} = \frac{\Omega^{1/2}}{\rho} \,, \qquad \Omega \equiv (\sigma_1 \kappa_1)^{\frac{1}{(1+a^2)}} \psi_1^{\frac{2a^2}{1+a^2}} \,. \tag{4.48}$$

In the patch described by the new coordinates, the solution takes the form

$$ds_\beta^2 = (\sigma'\kappa')^{\frac{2}{1+a^2}} \psi'^{\frac{4a^2}{1+a^2}} \left( d\rho'^2 + \rho'^2 d\Omega_{(2)}^2 \right) \,, \tag{4.49a}$$

$$E_{m'} = -\frac{2}{\sqrt{1+a^2}} e^{a\phi_\infty/2} e^{a\phi/2} \partial_{m'} \log\left(\sigma'/\kappa'\right) \,, \tag{4.49b}$$

$$e^{-a\phi} = e^{-a\phi_\infty} \left( \frac{\sigma_1 \kappa_1}{\psi_1^2} \right)^{\frac{2a^2}{1+a^2}} \left( \frac{\sigma'\kappa'}{\psi'^2} \right)^{\frac{2a^2}{1+a^2}} \,, \tag{4.49c}$$

where we have defined the functions

$$\psi' = 1 + \frac{\Omega/\psi_1}{\rho'} \,, \qquad \sigma' = 1 + \frac{\Omega/\sigma_1}{\rho'} \,, \qquad \kappa' = 1 + \frac{\Omega/\kappa_1}{\rho'} \,. \tag{4.50}$$

Thus, it is of the same form as in the other coordinate patch: asymptotically flat and with mass, electric and magnetic charges and asymptotic value of the dilaton given by

$$M' = \frac{1}{1+a^2} \Omega \left( \frac{1}{\sigma_1} + \frac{1}{\kappa_1} + 2a^2 \frac{1}{\psi_1} \right) \,, \tag{4.51a}$$

$$q' = q \,, \tag{4.51b}$$

$$\Sigma' = -\frac{2a}{1+a^2} \Omega \left( \frac{1}{\sigma_1} + \frac{1}{\kappa_1} - 2\frac{1}{\psi_1} \right) \tag{4.51c}$$

$$\phi'_\infty = \phi_\infty - \frac{2a}{1+a^2} \log\left(\frac{\sigma_1 \kappa_1}{\psi_1^2}\right). \tag{4.51d}$$

As in the JNW case, the mass and scalar charge take different values in the two asymptotic regions. The interaction energy $M_{int} = M - M'$ vanishes when

$$\psi_1^2 = \sigma_1 \kappa_1, \tag{4.52}$$

and the same condition makes the scalar charge (as well as the asymptotic value of the dilaton) equal in both regions. When this condition holds, the scalar charge is not independent anymore, but it is given in terms of the rest of the physical constants by

$$\Sigma \overset{a\neq 1}{=} -\frac{2a}{1-a^2}\left\{ M - \sqrt{M^2 - \tfrac{1}{4}(1-a^2)e^{a\phi_\infty}q^2} \right\}, \tag{4.53a}$$

$$\Sigma \overset{a=1}{=} -\frac{e^{\phi_\infty}q^2}{4M}. \tag{4.53b}$$

This is the value of the scalar charge in the complete, static, electric EMD solutions reviewed in Appendix A. In stationary spacetimes one can construct a definition of scalar charge which is coordinate-independent and satisfies a Gauss law [27, 28, 36]. The value of this charge for black holes with bifurcate horizons in EMD theories is precisely the one we have found by demanding the vanishing of the interaction energy. Furthermore, the fact that this value is the same in the two asymptotic regions is in agreement with the Gauss law satisfied by this kind of charge in stationary spacetimes. We conclude that, once more, vanishing interaction energy points to the existence of a stationary solution of which the initial data are a constant-time slice.

Notice that, in a given theory (*i.e.* for a given value of $a$) all scalar charges have the same sign, just as all masses do. That sign is purely conventional.

Let us now move to the $\rho_i \equiv |\vec{x} - \vec{x}_i| \to 0$ limit in the multi-center case. Defining the new coordinate

$$\frac{\rho'_i}{\Omega_i^{1/2}} = \frac{\Omega_i^{1/2}}{\rho_i}, \qquad \Omega_i \equiv (\sigma_i \kappa_i)^{\frac{1}{(1+a^2)}} \psi_i^{\frac{2a^2}{1+a^2}}, \tag{4.54}$$

the solution takes the following form in that limit

$$ds_\beta^2 \sim (\sigma'\kappa')^{\frac{2}{1+a^2}} \psi'^{\frac{4a^2}{1+a^2}} \left( d\rho_i'^2 + \rho_i'^2 d\Omega_{(2)i}^2 \right), \tag{4.55a}$$

$$E_{m'_i} \sim -\frac{2}{\sqrt{1+a^2}} e^{a\phi_\infty/2} e^{a\phi/2} \partial_{m'_i} \log\left(\sigma'/\kappa'\right), \qquad (4.55b)$$

$$e^{-a\phi} \sim e^{-a\phi_\infty} \left(\frac{\sigma_i \kappa_i}{\psi_i^2}\right)^{\frac{2a^2}{1+a^2}} \left(\frac{\sigma'\kappa'}{\psi'^2}\right)^{\frac{2a^2}{1+a^2}}, \qquad (4.55c)$$

where we have defined the functions

$$\psi' = 1 + \frac{A_i \Omega_i / \psi_1}{\rho'_i}, \qquad \sigma' = 1 + \frac{B_i \Omega_i / \sigma_i}{\rho'_i}, \qquad \kappa' = 1 + \frac{C_i \Omega_i / \kappa_i}{\rho'_i}, \qquad (4.56)$$

and the constants

$$A_i = 1 + \sum_{j \neq i} \frac{\psi_j}{|\vec{x}_{ij}|}, \qquad B_i = 1 + \sum_{j \neq i} \frac{\sigma_j}{|\vec{x}_{ij}|}, \qquad C_i = 1 + \sum_{j \neq i} \frac{\kappa_j}{|\vec{x}_{ij}|}. \qquad (4.57)$$

The values of the physical constants computed in the new asymptotic region are

$$M_i = \frac{1}{1+a^2} \Omega_i \left(\frac{B_i}{\sigma_i} + \frac{C_i}{\kappa_i} + 2a^2 \frac{A_i}{\psi_i}\right), \qquad (4.58a)$$

$$q_i = -\frac{2e^{-a\phi_\infty/2}}{\sqrt{1+a^2}} \Omega_i \left(\frac{\sigma_i \kappa_i}{\psi_i^2}\right)^{\frac{a^2}{1+a^2}} \left(\frac{B_i}{\sigma_i} - \frac{C_i}{\kappa_i}\right)$$

$$= -\frac{2e^{-a\phi_\infty/2}}{\sqrt{1+a^2}} \left(B_i \kappa_i - C_i \sigma_i\right), \qquad (4.58b)$$

$$\Sigma_i = -\frac{2a}{1+a^2} \Omega_i \left(\frac{B_i}{\sigma_i} + \frac{C_i}{\kappa_i} - 2\frac{A_i}{\psi_i}'\right) \qquad (4.58c)$$

$$\phi'_{\infty i} = \phi_\infty - \frac{2a}{1+a^2} \log\left(\frac{\sigma_i \kappa_i}{\psi_i^2}\right). \qquad (4.58d)$$

It is not difficult to see that the total electric charge is conserved

$$q = \sum_{i=1}^{N} q_i. \qquad (4.59)$$

Neither the mass nor the scalar charge enjoy the same property. In particular,

$$M_{int} \equiv M - \sum_{i=1}^{N} M_i$$

$$= \frac{1}{1+a^2} \sum_{i=1}^{N} \left[ \sigma_i + \kappa_i + 2a^2 \psi_i - \left( \frac{\sigma_i \kappa_i}{\psi_i^2} \right)^{-\frac{a^2}{1+a^2}} \left( B_i \kappa_i + C_i \sigma_i + 2a^2 A_i \psi_i \frac{\sigma_i \kappa_i}{\psi_i^2} \right) \right].$$
(4.60)

Our study of the single-center case indicates that part of the interaction energy can be understood as a property of each individual center and not to the interaction with other centers, *i.e.* as a sort of *self-interaction energy*. This part of the interaction energy can be removed by setting

$$\psi_i^2 = \sigma_i \kappa_i,$$
(4.61)

which, as can be easily checked, leads to the same asymptotic value for the dilaton in all the asymptotic regions and also imposes the relations Eqs. (4.53) between the physical parameters of the individual centers $\Sigma_i, M_i, q_i, \phi_\infty$.

Then ,

$$M_{int} = \frac{2}{1+a^2} \sum_{i \neq j} \frac{\sigma_i \kappa_j + 2a^2 (\sigma_i \kappa_j \sigma_j \kappa_i)^{1/2}}{|\vec{x}_{ij}|},$$
(4.62)

can be understood as the interaction energy between the non-extremal EMD black holes described by the well-known solutions described in Appendix A, which satisfy the relations Eqs. (4.53) and have no primary scalar hair.

In the $N = 2$ case we can express the interaction energy in terms of the physical constants of the individual centers as

$$M_{int} = -\frac{M_1 M_2}{|\vec{x}_{12}|} + e^{a\phi_\infty} \frac{q_1 q_2 / 4}{|\vec{x}_{12}|} - \frac{\Sigma_1 \Sigma_2 / 4}{|\vec{x}_{12}|} + \mathcal{O}(1/|\vec{x}_{12}|^2).$$
(4.63)

This is one of our main results. Observe that the interaction between the scalar charges of the centers is always attractive.

For identical centers with individual masses and charges $M, q, \Sigma$, the no-force condition (zero interaction energy) takes the form

$$M^2 - e^{a\phi_\infty} q^2 / 4 + \Sigma^2 / 4 = 0.$$
(4.64)

Since $\Sigma$ is the function of $M, q, \phi_\infty$ given in Eqs. (4.53), we can express the above condition as a relation between $M, q, \phi_\infty$ only. This relation should take the form of a

BPS bound. We find, for all non-vanishing values of $a$, the following relations for each of the centers:

$$M = \frac{e^{a\phi_\infty/2}}{2\sqrt{1+a^2}}|q|,\qquad(4.65)$$

so that

$$\Sigma \overset{a\neq 1}{=} -4aM,\qquad \Sigma \overset{a=1}{=} -2M.\qquad(4.66)$$

For these values of the physical parameters, the solutions are singular ($\psi_1 = -M/2 < 0$) when $a \neq 1,0$ and the result must be understood as a limit which, on the other hand, coincides with what is known of the extremal EMD black holes. The solutions are regular for $a = 0$ (ERN) and for $a = 1$.

Finally, to end this section we notice that for particular values of the parameters one obtains solutions which are cylindrical in the $\rho \to 0$ limit so that there is no second asymptotically-flat region. For a single center, this happens in the following cases:[11]

1. $\psi_1 = 0, \sigma_1 \neq 0, \kappa_1 \neq 0, a = \pm\sqrt{3}$.

2. $\psi_1 \neq 0, \sigma_1 = 0, \kappa_1 \neq 0, a = 0$, (and the $\sigma \leftrightarrow \kappa$ symmetric case).

3. $\psi_1 \neq 0, \sigma_1 = \kappa_1 = 0, a = \pm 1$.

## 4.6   Magnetic and dyonic EMD black holes

As in the Reissner–Nordström case, we can transform a purely electric into a purely magnetic solution using the discrete electric-magnetic duality transformations Eqs. (3.16). For $a \neq 0$, though, these transformations are not part of a continuous electric-magnetic duality group and one cannot generate dyonic from purely electric solutions with them. The purely magnetic solutions one obtains from the purely electric ones enjoy the same properties, upon the replacement of the electric charge by the magnetic charge. In particular, the solutions corresponding to extremal black holes are singular for all $a \neq 0, 1$.

Dyonic black hole solutions have been found in fully analytic form for the particular values $a = 1$ [31, 37] and $a = \sqrt{3}$ [38, 31, 39, 37, 40]. For other values of $a$ it has been shown that black-hole solutions exist, although with loss of analyticity in the horizon except for particular values of $a$ [41, 42].

In what follows, we are going to present and study new solutions to the time-symmetric initial data problem for the case $a = 1$. These are given in terms of 4 harmonic functions in $\mathbb{E}^3$, $\sigma, \kappa, \psi, \tau$,

---

[11]In the cases in which $\Phi^4 \sim \rho^{-3}$ or $\rho^{-1}$ in the $\rho \to 0$ limit, the analytical extension gives rise to an asymptotically-conical spacetime and we will not study them here.

$$\partial_m \partial_m (\psi, \sigma, \psi, \tau) = 0 \,, \tag{4.67}$$

by

$$\Phi^4 = \sigma \kappa \psi \tau \tag{4.68a}$$

$$E_m = \alpha \sqrt{2} e^{\phi/2} \partial_m \log (\sigma/\kappa) \,, \tag{4.68b}$$

$$B_m = \beta \sqrt{2} e^{-\phi/2} \partial_m \log (\psi/\tau) \,, \tag{4.68c}$$

$$e^{-\phi} = e^{-\phi_\infty} \frac{\sigma \kappa}{\psi \tau} \,, \tag{4.68d}$$

where $\alpha$ and $\beta$ square to one and indicate the sign of all the electric and magnetic charges, respectively.

Asymptotically-flat solutions are obtained by choosing harmonic functions of the form (in the obvious notation)

$$\sigma, \kappa, \psi, \tau = 1 + \sum_{i=1}^{N} \frac{\sigma_i, \kappa_i, \psi_i, \tau_i}{|\vec{x} - \vec{x}_i|} \,, \tag{4.69}$$

where all the coefficients $\sigma_i, \kappa_i, \psi_i, \tau_i$ are assumed to be non-negative.

Setting $\tau = \psi$ we recover the purely electric $a = 1$ solutions studied in Section 4.5 and with $\sigma \kappa = \psi \tau$ we recover the Reissner–Nordström ones of Sections 4.3 and 4.4.

### 4.6.1 Single-center case

In the $N = 1$ case, the mass $M$, electric and magnetic charges $q, p$ and scalar charge $\Sigma$ are given by[12]

$$M = \tfrac{1}{2} (\sigma_1 + \kappa_1 + \psi_1 + \tau_1) \,, \tag{4.71a}$$

$$q = \alpha \sqrt{2} e^{-\phi_\infty/2} (\sigma_1 - \kappa_1) \,, \tag{4.71b}$$

---

[12]The magnetic charge can be defined in a coordinate-invariant way by

$$p = \frac{1}{4\pi} \int_{S^2} d^2 \Sigma_m e^{\phi} B^m = -\rho^2 \Phi^2 e^{\phi} B_\rho \,. \tag{4.70}$$

Since this charge satisfies a Gauss law, this expression must be $\rho$-independent and the result of the integral should be the same for any 2-sphere centered in the origin.

$$p = \beta\sqrt{2}e^{\phi_\infty/2}(\psi_1 - \tau_1) \,, \tag{4.71c}$$

$$\Sigma = -\sigma_1 - \kappa_1 + \psi_1 + \tau_1 \,. \tag{4.71d}$$

Since the 4 integration constants $\sigma_1, \kappa_1, \psi_1, \tau_1$ are independent, the above 4 charges, $M, q, p, \Sigma$ are also independent and the solution has primary scalar hair.

When the 4 integration constants $\sigma_1, \kappa_1, \psi_1, \tau_1$ are all non-vanishing, there is another asymptotically-flat region in the $\rho_1 \equiv |\vec{x} - \vec{x}_1| \to 0$ limit. Using the same methods we have employed in the previous cases, we find that the mass and scalar charge of the solution and the asymptotic value of the dilaton in that asymptotic region are given by

$$M' = \tfrac{1}{2}\Omega \left( \frac{1}{\sigma_1} + \frac{1}{\kappa_1} + \frac{1}{\psi_1} + \frac{1}{\tau_1} \right) \,, \tag{4.72a}$$

$$\Sigma' = \Omega \left( -\frac{1}{\sigma_1} - \frac{1}{\kappa_1} + \frac{1}{\psi_1} + \frac{1}{\tau_1} \right) \,, \tag{4.72b}$$

$$\phi'_\infty = \phi_\infty + \log \left( \frac{\psi_1\tau_1}{\sigma_1\kappa_1} \right) \,, \tag{4.72c}$$

with

$$\Omega^2 \equiv \sigma_1\kappa_1\psi_1\tau_1 \,, \tag{4.73}$$

while the electric and magnetic charges are equal because they satisfy Gauss laws.

The interaction mass $M_{int} = M - M'$ is generically different from zero. It vanishes when the product of any two integration constants equals the product of the other two. For the sake of concreteness we are going to choose

$$\sigma_1\kappa_1 = \psi_1\tau_1 \,. \tag{4.74}$$

It is not difficult to see that, as in all previous cases, when the interaction energy vanishes, the scalar charge and the asymptotic value of the scalar take the same values in both asymptotic regions $\Sigma = \Sigma'$, $\phi'_\infty = \phi_\infty$. Furthermore, the above relation between the 4 integration constants implies the existence of a relation between the 4 charges that allows us to express the scalar charge in terms of the conserved charges

$$\Sigma = \frac{e^{-\phi_\infty}p^2 - e^{\phi_\infty}q^2}{4M} \,, \tag{4.75}$$

which indicates the absence of primary scalar charge. These solutions are constant-time slices of the static solutions constructed in Refs. [31, 37]. When the constraint

Eq. (4.74) is not imposed, the initial data describe a dynamical black hole with primary scalar hair.

When only two of the integration constants are non-vanishing, the solution is asymptotically cylindrical in the $\rho_1 \to 0$ limit. Observe that, in this case, there is always a (vanishing) product of two integration constants which is equal to the (vanishing) product of the other two. This leads to vanishing interaction energy and equal scalar charges and asymptotic values of the scalar in both asymptotic regions. If we take, for instance, $\kappa_1 = \tau_1 = 0$, it is easy to see that the solutions are constant-time slices of extremal, static, black holes saturating the bound

$$M = \frac{e^{\phi_\infty/2}}{2\sqrt{2}}|q| + \frac{e^{-\phi_\infty/2}}{2\sqrt{2}}|p|\,, \tag{4.76a}$$

$$\Sigma = -\frac{e^{\phi_\infty/2}}{2\sqrt{2}}|q| + \frac{e^{-\phi_\infty/2}}{2\sqrt{2}}|p|\,. \tag{4.76b}$$

### 4.6.2 The multi-center case

For general values of $N$, we find the total charges

$$M = \tfrac{1}{2}\sum_{i=1}^{N}\left(\sigma_i + \kappa_i + \psi_i + \tau_i\right)\,, \tag{4.77a}$$

$$q = \alpha\sqrt{2}e^{-\phi_\infty/2}\sum_{i=1}^{N}(\sigma_i - \kappa_i)\,, \tag{4.77b}$$

$$p = \beta\sqrt{2}e^{\phi_\infty/2}\sum_{i=1}^{N}(\psi_i - \tau_i)\,, \tag{4.77c}$$

$$\Sigma = \sum_{i=1}^{N}\left(-\sigma_i - \kappa_i + \psi_i + \tau_i\right)\,. \tag{4.77d}$$

When all the integration constants are different from zero there is another asymptotically-flat region in each $\rho_i \equiv |\vec{x} - \vec{x}_i| \to 0$ limit in which the charges and the asymptotic values of the scalar take the values

$$M_i = \tfrac{1}{2}\left(\sigma_i' + \kappa_i' + \psi_i' + \tau_i'\right)\,, \tag{4.78a}$$

$$q_i = -\alpha\sqrt{2}e^{-\phi_{\infty i}/2}(\sigma_i' - \kappa_i')\,, \tag{4.78b}$$

$$p_i = -\beta\sqrt{2}e^{\phi_{\infty i}/2}(\psi_i' - \tau_i')\,, \tag{4.78c}$$

$$\Sigma_i = -\sigma_i' - \kappa_i' + \psi_i' + \tau_i'{}_{,,} \tag{4.78d}$$

$$\phi_{\infty i} = \phi_\infty + \log\left(\frac{\psi_i \tau_i}{\sigma_i \kappa_i}\right)\,, \tag{4.78e}$$

with

$$\sigma_i' \equiv \Omega_i\left(1 + \sum_{j\neq i}^N \frac{\sigma_j}{|\vec{x}_{ij}|}\right)/\sigma_i\,, \qquad \Omega_i^2 \equiv \sigma_i \kappa_i \psi_i \tau_i\,, \tag{4.79}$$

and similar expressions for $\kappa_i', \psi_i', \tau_i'$.

It is easy to check that the sum of the individual electric and magnetic charges give the total electric and magnetic charge computed in the asymptotic region that contains all the centers

$$\sum_{i=1}^N q_i = q\,, \qquad \sum_{i=1}^N p_i = p\,, \tag{4.80}$$

as they should. As usual, the same is not true for the individual masses $M_i$ and the total mass $M$ nor for the scalar charges. In order to study the interaction between the different black holes, we must remove the self-interaction due to the presence of primary scalar hair, using the condition Eq. (4.74) for each center, *i.e.*, imposing

$$\sigma_i \kappa_i = \psi_i \tau_i\,, \quad \forall i = 1, \ldots, N\,. \tag{4.81}$$

This condition makes all the asymptotic values of the scalar equal and, furthermore, the scalar charge satisfies a Gauss law

$$\sum_{i=1}^N \Sigma_i = \Sigma\,. \tag{4.82}$$

The remaining interaction energy has the simple, exact, expression

$$M_{int} = M - \sum_{i=1}^N M_i = -\sum_{i,j\neq i}^N \frac{\sigma_j \kappa_i + \kappa_j \sigma_i + \psi_j \tau_i + \tau_j \psi_i}{|\vec{x}_{ij}|}\,, \tag{4.83}$$

which can be rewritten in terms of the charges of two centers in the following approximate way

$$M_{int} = -\frac{M_1 M_2}{|\vec{x}_{12}|} + \frac{e^{\phi_\infty}}{4}\frac{q_1 q_2}{|\vec{x}_{12}|} + \frac{e^{-\phi_\infty}}{4}\frac{p_1 p_2}{|\vec{x}_{12}|} - \frac{1}{4}\frac{\Sigma_1 \Sigma_2}{|\vec{x}_{12}|} + \mathcal{O}\left(\frac{1}{|\vec{x}_{12}|^2}\right), \quad (4.84)$$

which is commonly used in the literature for static, widely separated black holes.

For identical black holes, the vanishing of the interaction energy leads to the well-known BPS-type relation

$$M^2 - \frac{e^{\phi_\infty} q^2}{4} - \frac{e^{-\phi_\infty} p^2}{4} + \frac{1}{4}\Sigma^2 = 0. \quad (4.85)$$

Since we have shown that the no-primary-hair condition implies Eq. (4.75), it is not difficult to use this relation to rewrite the above condition as

$$\left[M - \tfrac{1}{2\sqrt{2}}\left(e^{\phi_\infty/2}|q| + e^{-\phi_\infty/2}|p|\right)\right]\left[M - \tfrac{1}{2\sqrt{2}}\left(e^{\phi_\infty/2}|q| - e^{-\phi_\infty/2}|p|\right)\right] = 0, \quad (4.86)$$

which shows that the condition is satisfied when the mass equals either of the two central charge skew eigenvalues[13]

$$\mathcal{Z}_\pm \equiv \tfrac{1}{2\sqrt{2}}\left(e^{\phi_\infty/2}|q| \pm e^{-\phi_\infty/2}|p|\right). \quad (4.87)$$

It is not difficult to see that, in these cases, at least two integration coefficients of the center must vanish and that in the $\rho_i \to 0$ limit the solution becomes cylindrical, corresponding to the infinite throat of an extremal black hole.

# 5   Discussion

In this paper we have shown how our interpretation of the non-vanishing *self-interaction energy* is related to the existence of primary scalar hair and to the necessary evolution of the black hole solution. We would like to stress that the black-hole initial data with primary scalar hair are completely regular. However this does not contradict the no-hair conjecture, which only refers to the endpoint of gravitational collapse and therefore, to the stationary state which will be eventually reached.

This raises some important questions that we have already posed in the previous sections and which fall out of the scope of this paper: will the final state of the evolution of the black holes with primary hair contain some primary hair or will it be completely lost (radiated away)? Will the final state be a regular black hole?

Studying the evolution of the initial data describing single black holes with primary hair which we have provided should be possible using numerical methods. We believe there is a great deal of black-hole physics to be learned from this problem.

---

[13]These combinations have that interpretation when the $a = 1$ theory is embedded in pure $\mathcal{N} = 4, d = 4$ ungauged supergravity. See Refs. [43–46].

# Acknowledgments

UKBV would like to thank Álvaro de la Cruz–Dombriz for his support and acknowledge the hospitality of the IFT-UAM/CSIC in Madrid during the earliest stages in the preparation of the manuscript. TOM would also like to thank Álvaro de la Cruz–Dombriz for bringing him the opportunity to participate in the Erasums+ program. UKBV acknowledges financial support from the National Research Foundation of South Africa, Grant number PMDS22063029733, from the University of Cape Town Postgraduate Funding Office, the University of Groningen and the Erasmus+ KA107 International Credit Mobility Programme. This work has been supported in part by the MCI, AEI, FEDER (UE) grants PID2021-125700NB-C21 ("Gravity, Supergravity and Superstrings" (GRASS)), and IFT Centro de Excelencia Severo Ochoa CEX2020-001007-S. TO wishes to thank M.M. Fernández for her permanent support.

# A Static, electric, black-hole solutions of the EMD model

The static, purely electric, black-hole solutions[14] of the EMD model presented in Eq. (2.1) were found in Refs. [31–33] and can be written in the convenient form

$$ds^2 = H^{-\frac{2}{1+a^2}} W dt^2 - H^{\frac{2}{1+a^2}} \left[ W^{-1} dr^2 + r^2 d\Omega^2_{(2)} \right] , \tag{A.1a}$$

$$A_t = \alpha e^{a\phi_\infty/2}(H^{-1} - 1) , \tag{A.1b}$$

$$e^{-\phi} = e^{-\phi_\infty} H^{\frac{2a}{1+a^2}} , \tag{A.1c}$$

where the functions $H$ and $W$ (often called the *blackening factor*) take the form

$$H = 1 + \frac{h}{r} , \qquad W = 1 + \frac{\omega}{r} , \tag{A.2}$$

and where the integration constants $h, \omega, \alpha$ are related by

$$\omega = h\left[1 - (1 + a^2)(\alpha/2)^2\right] . \tag{A.3}$$

The physical parameters of these solutions are the ADM mass $M$, the electric charge $q$ and the asymptotic value of the scalar (*modulus*) $\phi_\infty$. The integration constants are given in terms of them by

---

[14]Solutions including primary scalar hair were found in Ref. [47]. They are singular when the primery hair is present.

$$h = -\frac{a^2+1}{a^2-1}\left\{ M - \sqrt{M^2 + \tfrac{1}{4}(a^2-1)e^{a\phi_\infty}q^2} \right\},$$

$$\omega = -\frac{2}{a^2-1}\left\{ a^2 M - \sqrt{M^2 + \tfrac{1}{4}(a^2-1)e^{a\phi_\infty}q^2} \right\}, \qquad \text{(A.4)}$$

$$\alpha = -4qe^{a\phi_\infty/2}/h,$$

for $a \neq 1$ and

$$h = \frac{4e^{\phi_\infty}q^2}{M},$$

$$\omega = -2\frac{M^2 - 2e^{\phi_\infty}q^2}{M}, \qquad \text{(A.5)}$$

$$\alpha = -e^{-\phi_\infty/2}M/q,$$

for $a = 1$.

The scalar charge $\Sigma$, conventionally defined by

$$\phi \sim \phi_\infty + \frac{\Sigma}{r}, \qquad \text{(A.6)}$$

takes the value

$$\Sigma = -\frac{2ah}{a^2+1}, \qquad \text{(A.7)}$$

and we can write the integration constant $\omega$ in terms of the physical constants $M$ and $\Sigma$ as

$$\omega = -\frac{1}{a}\left[2aM - \Sigma\right]. \qquad \text{(A.8)}$$

We have chosen the sign of the square roots in $h$ and $\omega$ so that $h$ is always positive and $\omega$ is always negative when the non-extremality condition

$$M^2 > \frac{4}{a^2+1}e^{a\phi_\infty}q^2, \qquad \text{(A.9)}$$

is met. In that case, there is an event horizon at

$$r = -\omega \equiv r_0. \qquad \text{(A.10)}$$

Its Bekenstein-Hawking entropy and Hawking temperature are given by

$$S = \pi r_0^{\frac{2a^2}{a^2+1}} (r_0 + h)^{\frac{2}{a^2+1}}, \tag{A.11}$$

$$T = \frac{r_0}{4S}. \tag{A.12}$$

In the extremal limit ($\omega = r_0 = 0$) all the solutions become singular except for $a = 0$.

# B Isotropic coordinates

Often, we find 3-dimensional metrics conformal to one of the form

$$W^{-1}dr^2 + r^2 d\Omega_{(2)}^2, \qquad W = \left(1 + \frac{\omega}{r}\right). \tag{B.1}$$

The coordinate transformation

$$r = \rho \left(1 - \frac{\omega/4}{\rho}\right)^2, \tag{B.2}$$

brings it to the form

$$\left(1 - \frac{\omega/4}{\rho}\right)^4 \left(d\rho^2 + \rho^2 d\Omega_{(2)}^2\right). \tag{B.3}$$

## B.1 The Schwarzschild solution in isotropic coordinates

Making this coordinate transformation in the Schwarzschild metric

$$ds^2 = \left(1 - \frac{2M}{r}\right) dt^2 - \left(1 - \frac{2M}{r}\right)^{-1} dr^2 - r^2 d\Omega_{(2)}^2, \tag{B.4}$$

($\omega = -2M$), it takes the form

$$ds^2 = \frac{\left(1 - \frac{M/2}{\rho}\right)^2}{\left(1 + \frac{M/2}{\rho}\right)^2} dt^2 - \left(1 + \frac{M/2}{\rho}\right)^4 \left(d\rho^2 + \rho^2 d\Omega_{(2)}^2\right). \tag{B.5}$$

## B.2 The Janis-Newman-Winicour solution in isotropic coordinates

Upon the transformation Eq. (B.2), the Janis–Newman–Winicour (JNW) solution [30]

$$ds^2 = W^{-\alpha}dt^2 - W^{1+\alpha}\left[W^{-1}dr^2 + r^2 d\Omega^2_{(2)}\right], \tag{B.6a}$$

$$\phi = \phi_0 \pm \sqrt{1-\alpha^2}\log W, \tag{B.6b}$$

takes the form

$$ds^2 = (\psi/\chi)^{-2\alpha}dt^2 - (\psi\chi)^2(\psi/\chi)^{2\alpha}\left(d\rho^2 + \rho^2 d\Omega^2_{(2)}\right), \tag{B.7a}$$

$$\phi = \phi_0 \pm 2\sqrt{1-\alpha^2}\log(\psi/\chi), \tag{B.7b}$$

with

$$\psi = 1 + \frac{\omega/4}{\rho}, \qquad \chi = 1 - \frac{\omega/4}{\rho}. \tag{B.8}$$

## B.3 The electric Einstein–Maxwell–Dilaton solution in isotropic coordinates

Performing the same transformation in the electric EMD solutions in Eqs. (A.1) they are brought to the form

$$ds^2 = (\sigma\kappa)^{-\frac{2}{1+a^2}}\psi^{-2+\frac{4}{1+a^2}}\chi^2 dt^2 - (\sigma\kappa)^{\frac{2}{1+a^2}}\psi^{\frac{4a^2}{1+a^2}}\left(d\rho^2 + \rho^2 d\Omega^2_{(2)}\right), \tag{B.9a}$$

$$A_t = \alpha e^{a\phi_\infty/2}\left(\frac{\psi^2}{\sigma\kappa} - 1\right), \tag{B.9b}$$

$$\phi = \phi_\infty - \frac{2a}{1+a^2}\log\left(\frac{\sigma\kappa}{\psi^2}\right), \tag{B.9c}$$

with

$$\psi = 1 + \frac{\psi_1}{\rho}, \qquad \chi = 1 + \frac{\chi_1}{\rho}, \qquad \sigma = 1 + \frac{\sigma_1}{\rho}, \qquad \kappa = 1 + \frac{\kappa_1}{\rho}, \tag{B.10}$$

and

$$\psi_1 = -\omega/4, \qquad\qquad \chi_1 = \omega/4,$$

$$\sigma_1 = \frac{\omega^2}{4h\left(1 + \alpha\sqrt{1+a^2}/2\right)^2}, \qquad \kappa_1 = \frac{\omega^2}{4h\left(1 - \alpha\sqrt{1+a^2}/2\right)^2}, \tag{B.11}$$

where the constants $\omega, h, \alpha$ are related by the constraint Eq. (A.3).

Observe that

$$\psi_1 = -\chi_1, \qquad \frac{\psi_1^2}{\sigma_1 \kappa_1} = 1. \tag{B.12}$$

These solutions include the Reissner-Nordström one for $a = 0$. It takes the form

$$ds^2 = \left(\frac{\psi\chi}{\sigma\kappa}\right)^2 dt^2 - (\sigma\kappa)^2 \left(d\rho^2 + \rho^2 d\Omega_{(2)}^2\right), \tag{B.13a}$$

$$A_t = \alpha\left(\frac{\psi^2}{\sigma\kappa} - 1\right), \tag{B.13b}$$

with

$$\psi_1 = -\omega/4, \qquad\qquad \chi_1 = \omega/4,$$

$$\sigma_1 = \frac{\omega^2}{4h\left(1 + \alpha/2\right)^2}, \qquad \kappa_1 = \frac{\omega^2}{4h\left(1 - \alpha/2\right)^2}. \tag{B.14}$$

For $\alpha = 0$ one recovers the Schwarzschild solution which has $\sigma = \kappa$.

This famiy of solutions does not include the JNW one. A more general solution such as the Agnese-La Camera solution [47] is needed to cover all the possiblities.

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
