# Peer review of "On the interactions and equilibrium between Einstein-Maxwell-Dilaton black holes"

_SciPost Physics Core_

## Round 1 · Author Response

These are our answers to the referee's questions and comments:
1.- We believe there is a misunderstanding here. What we study in this paper
is the interaction between black holes which are in the same universe,
building on the classical work of Brill and Lindquist in the
Einstein-Maxwell theory and introducing some new ideas that allow us to
determine the equilibrium conditions between black holes in the
Einstein-Maxwell-Dilaton theory.
Time-symmetric initial data that describe black-hole solutions have a long
history (see the references in the manuscript). For this particular case, it
turns out that one can find initial data that describe more than one black
hole (always in the same universe). In all cases, (one or many black holes)
one can analytically extend the solutions finding Einstein-Rosen bridges
that connect each black hole to another asymptotically-flat region (that we
may call universe). The presence of those bridges does not modify the fact
that there is a universe in which all the original black holes are present,
interacting exactly in the same fashion. However, in the universes on the
other side of the Einstein-Rosen bridge are different for each black
hole. Technically, this allows us to compute the black holes' individual
masses, because in those universes the black holes are not interacting with
other black holes (this is Brill and Lindquist's idea).
The physics of these systems is quite clear and simple. One may object to
the Einstein-Rosen-like bridges, but their existence simply unavoidable. The
solutions have them. In astrophysically relevant black holes, formed by
gravitational collapse, they do not exist, certainly. But
Einstein-Maxwell-Dilaton (or Reissner-Nordstr\"om) black holes do not exist
in the universe nor are created by gravitational collapse of known stars,
either. However, as we explain in the (new) third paragraph of the
introduction, these initial data (with their Einstein-Rosen bridges) can and
have actually been used to study head-on black hole collisions. Thus, they
have been considered by experts as physically viable, even if for very
restricted cases.
Our interest on those black holes and our construction are purely
theoretical. We are interested in determining the forces and equilibrium
conditions between the black holes of that theory. We also manage to
distinguish between the cases which will evolve and those which are part of
a static solution. The results obtained are consistent.
Since we are no experts in numerical relativity, we cannot comment on
numerical initial data and their evolution. However, we stress that our
solutions (just like Brill and Lindquist's or Gibbons, Cveti\v{c} and
Pope's) are just initial data given in analytical form. This is perhaps
unusual because the initial data problem is complicated and only numerical
solutions are usually found. The beauty of the time-symmetric case is that
it can be solved analytically. The drawback is that time-symmetric initial
data can only describe head-on black-hole collisions. We have added a long
paragraph to the introduction (paragraph 3 in the revised version) with
references to the use of this kind of initial data to study head-on
Schwarschild black-hole solutions. Nothing forbids their use to study
head-on collisions of Einstein-Maxwell-Dilaton black holes.
Along the paper we compute the physical observables that can be computed in
this case in which the black holes have zero velocities, like masses and
charges of different kinds.
2.- We have added the references suggested by the referee to the footnote~2
within a new paragraph.
The referee considers our manuscript as too technical and mathematically
heavy. However, the mathematics we have used are a generalization of those
used by Misner, Brill and Lindquist more than 60 years ago. The solutions
found are simple generalizations with a few more functions and parameters and
their analytical extension is completely standard (again, the same used on
those classical papers). The use of arguments based on the existence of
charges satisfying Gauss laws, which is one of the novelties of our
manuscript, cannot be considered, in our opinion, as too advanced. Finally,
the results obtained are physical: forces and equilibrium conditions between
Einstein-Maxwell-Dilaton black holes.
Thus, we do not believe our manuscript is purely technical and we do not feel
it is inappropriate for SciPost Core, although we will respect the editor's
decision in this respect.

---

## Round 1 · List of Changes

1.- We have added a long paragraph to the introduction (paragraph 3 in the revised version) with
references to the use of this kind of initial data to study head-on Schwarschild black-hole solutions.
2.- We have added the references suggested by the referee to the footnote~2
within a new paragraph.

---

## Editorial Decision

editorial_decision: